# Disease parameters following ocular herpes simplex virus type 1 infection are similar in male and female BALB/C mice

**Aaron W. Kolb[1], Sarah A. Ferguson[1], Inna V. Larsen[1], Curtis R. Brandt**[1,2,3]\*

**1** Department of Ophthalmology and Visual Sciences, University of Wisconsin-Madison School of Medicine and Public Health, Madison, WI, United States of America, **2** Department of Medical Microbiology and Immunology, University of Wisconsin-Madison School of Medicine and Public Health, Madison, WI, United States of America, **3** McPherson Eye Research Institute, University of Wisconsin-Madison, Madison, WI, United States of America

\* crbrandt@wisc.edu

**Data Availability Statement:** The sequence alignment consisting of the 20 recombinants, parental strains and strain 17 as well as raw ocular

## Abstract

Sex related differences in the incidence or severity of infection have been described for multiple viruses. With herpes simplex viruses, the best example is HSV-2 genital infection where women have a higher incidence of infection and can have more severe infections than men. HSV-1 causes several types of infections including skin and mucosal ulcers, keratitis, and encephalitis in humans that do not appear to have a strong biological sex component. Given that mouse strains differ in their MHC loci it is important to determine if sex differences occur in multiple strains of mice. Our goal was to answer two questions: Are virus related sex differences present in BALB/C mice and does virulence of the viral strain have an effect? We generated a panel of recombinant HSV-1 viruses with differing virulence phenotypes and characterized multiple clinical correlates of ocular infection in BALB/c mice. We found no sex-specific differences in blepharitis, corneal clouding, neurovirulence, and viral titers in eye washes. Sex differences in neovascularization, weight loss and eyewash titers were observed for some recombinants, but these were not consistent across the phenotypes tested for any recombinant virus. Considering these findings, we conclude that there are no significant sex specific ocular pathologies in the parameters measured, regardless of the virulence phenotype following ocular infection in BALB/c mice, suggesting that the use of both sexes is not necessary for the bulk of ocular infection studies.

## Introduction

Biological sex specific differences for both incidence of symptomatic disease and disease severity have been described for several viruses including herpes viruses, dengue virus, hantaviruses, hepatitis viruses, HIV, HTLV-1, influenza virus, measles, West Nile virus, encephalomyocarditis virus, vesicular stomatitis virus, coxsackie virus 3B [1–16]. Some of these studies have been done at the organismal level and others at the cell and molecular level where sex differences seem to depend on the outcome measure or model system used. Generally, females are thought

disease and neurovirulence scores can be found at the URL https://doi.org/10.5061/dryad.q83bk3jnw.

**Funding:** This work was funded by National Institute of Health (www.nih.gov) grant R21AI137280 to CRB, a Vision Core Research grant from the National Eye Institute (www.nei.nih.gov/), P30EY016665 to CRB, and an unrestricted grant to the Department of Ophthalmology and Visual Sciences from Research to Prevent Blindness, Inc (www.rpbusa.org/rpb/). The funders had no role in study design, data collection and analysis, decision to publish, or preparation of the manuscript.

**Competing interests:** The authors have declared that no competing interests exist.

to be more resistant to virus induced pathologies, but this is not consistent across all viral infections. For SARS CoV-2, males are more susceptible to severe infections and some of this susceptibility may be related to the expression of innate immune genes located on the X chromosome [17–22].

For herpes simplex viruses some disease manifestations show a sex-specific difference while others do not. For example, in HSV-2 genital infections, women have a higher acquisition rate, higher incidence of symptoms, and greater prevalence than men [3, 23–28]. Differences related to steroid regulation of immune functions and physiology may be involved in HSV-2 genital infections [13]. Suligoi et. al. [29] reported sex differences in the incidence of HSV-1 infection for adolescents, however, other studies reported no sex differences in the incidence of HSV-2 infections in adolescents, or mouse peritoneal macrophages [13, 30].

Sex differences have also been reported in central nervous system infection with HSV-1 and VZV [31, 32] and androgens have been reported to play a role in differential disease severity in animal models of HSV-1 infection [33–35]. In IFN-γ knockout mice, sex specific differences have also been reported [36]. Gender specific differences in susceptibility to HSV-1 infection following exercise stress have also been reported [37]. In contrast, sex differences were not associated with the prevalence of HSV-1 DNA in human trigeminal ganglia [38]. Thus, reported sex differences for HSV-1 infections are not consistent.

Herpes simplex virus type 1 is a frequent cause of blindness due to infectious disease in developed countries [39, 40]. Herpetic keratitis is characterized by the initial development of conjunctivitis and corneal epithelial lesions. This triggers an inflammatory response with infiltration of neutrophils early on followed by migration of CD4+ T-cells, and other cells into the cornea. Primary infections in humans usually do not result in severe damage to the cornea, but multiple viral reactivation events can result in corneal edema, corneal clouding, and neovascularization of the cornea [41–47]. Thus, HSV-1 keratitis is an immunopathological disease. Epidemiological studies in humans have found no sex differences in the incidence of HSV-1 induced keratitis [40]. For other studies, sexes were not analyzed separately [48–51]. A previous study using NIH/OLA inbred mice found that eye disease followed a similar pattern in males and females [52]. Recently, studies in C57BL/6 mice using an ocular infection model with a single viral strain reported there were no significant sex differences in disease severity or immune cell infiltration of the cornea [53]. A study to map host cell genes that affect HSV-1 ocular infection using BXD mice also found no consistent sex differences [54].

The genetic background of different inbred mouse strains has been shown to affect corneal disease severity in HSV-1 ocular infections [2, 55, 56]. The BALB/c inbred mouse strain is widely used in ocular HSV-1 studies [57–61], and is more susceptible to HSV-1 corneal disease than C57BL/6 [2, 55, 56]. While biological sex differences do not appear to be a factor in C57BL/6 mice HSV-1 ocular infection, it is currently unclear whether the same is true in BALB/c [53], due to phenotypic differences in ocular disease severity between the two animal strains. Two critical questions remaining are whether sex differences exist in ocular HSV-1 infections in BALB/c mice and whether sex affects the virulence of a particular viral strain?

We previously described a novel approach for mapping virulence genes in HSV-1 infections termed virulence Quantitative Trait Locus Mapping (vQTLmap) and applied it to HSV-1 ocular infections in our BALB/C mouse model [62]. At that time, both sexes were not required to be included in studies, so those studies were done only in female mice. We have now isolated additional recombinant viruses with a wide range of ocular virulence and tested the disease phenotypes in both male and female BALB/c mice allowing us to compare susceptibility to ocular disease. We found that in general, there were no significant sex differences related to ocular disease. We observed some sex-specific phenotypes with a particular recombinant virus, but those differences were not consistent across all ocular disease phenotypes. Our

results are consistent with previous reports [53, 54] indicating that sex differences are not significantly involved in disease severity using ocular HSV-1 infection models in multiple strains of mice.

## Materials and methods

### Ethics statement

This study was carried out in strict accordance with the recommendations in the Guide for the Care and Use of Laboratory Animals of the National Institutes of Health. The protocol was approved by the Institutional Animal Care and Use Committee of the University of Wisconsin (Protocol Number: M006440). All corneal scarification, disease scoring, tear film collection was performed under isoflurane anesthesia. Sustained release buprenorphine was used for analgesia. Euthanasia was carried out by first anesthetizing the mice with isoflurane and then performing cervical dislocation.

### Cells

Vero cells (CCL-81; ATCC, Manassas, VA, USA) were used for producing viral stocks generating viral DNA. The cells were propagated in Dulbecco's modified Eagles medium (DMEM), supplemented with 5% serum (1:1 ratio of bovine calf and fetal bovine serum) plus antibiotics and grown at 37˚C with 5% $CO_2$.

### Viruses

The HSV-1 viruses in this study were recombinants generated by mixed corneal infection with two, avirulent, plaque purified clinical isolates, OD4 and CJ994, in 3- to 4-week-old BALB/c mice (Envigo, Indianapolis, IN, USA). The ocular virulence phenotypes of the parental OD4 and CJ994 strains have been previously described [63]. Corneal scarification was carried out using a 30-gauge needle followed by application of $1x10^5$ PFU of virus (1:1 ratio of OD4 and CJ994) in 5 μL of DMEM with 2% serum. Seven days following infection, the animals were sacrificed, and the trigeminal ganglia were removed, followed by tissue bead homogenization (CK14, Precellys, Rockville, MD, USA). The homogenate was subjected to 3 rounds of freeze-thaw, serially diluted, and plated on confluent Vero cell, 6-well plates for plaque purification. Each recombinant virus was plaque purified 3 times prior to the preparation of frozen stocks.

### Viral DNA purification and screening for recombinants

Viral genomic DNA was isolated from each recombinant and parent as previously described [62, 64]. Briefly, five confluent TC100 plates of Vero cells were infected with recombinant viral stock in DMEM + 2% serum. The infected cells were harvested 24 hours after the monolayer reached 100% cytopathic effect (CPE). The cells were centrifuged at 600 x g for 10 minutes, and the cell pellet with 5 mL of supernatant was subjected to three freeze-thaw cycles. The lysate was then combined with the remaining supernatant and centrifuged at 600 x g for 10 minutes. The supernatant was then layered on a 36% sucrose cushion (in phosphate buffered saline; PBS), and centrifuged for 80 min at $24,000 \times g$. Following centrifugation, the supernatant was removed, and the pellet was resuspended in 3 mL of TE (10 mM Tris [pH 7.4], 1 mM EDTA) buffer plus 0.15M sodium acetate (pH 5.5). The virus preparation was then incubated with 50 μg/μL of RNase A for 30 minutes at 37˚C. Proteinase K and SDS (50 μg/μL and 0.1% respectively) were then added and the preparation was incubated 30 minutes at 37˚C. The DNA was then purified by phenol/chloroform extraction. The DNA was then precipitated using ice-cold 95% ethanol and desalted with 70% ethanol, followed by resuspension in sterile

water. To determine if a strain was a recombinant, BamHI (R6021; Promega, Madison, WI, USA) digested DNA was electrophoresed on a 1% agarose-tris/borate/EDTA (TBE) gel, and the restriction fragment patterns were compared to those of the OD4 and CJ994 parental strains. A strain was determined to be a recombinant if the RFLP mapping revealed patterns which were a combination of the parental strains. Following confirmation as a recombinant by RFLP analysis, high titer stocks were produced as described previously [63].

## Genomic sequencing

Prior to genomic sequencing, the quality of the DNA from each sample was measured using a NanoDrop One (ThermoFisher Scientific, Waltham, MA, USA). Quantification of the extracted DNA was determined using a Qubit dsDNA High Sensitivity kit (ThermoFisher Scientific). The DNA samples were then diluted and loaded into an Agilent FemtoPulse (Agilent, Santa Clara, CA, USA) electrophoresis system to evaluate DNA size and quality. The samples were subsequently prepared as a Pacific Biosciences Microbial Multiplex library according to PN 101-696-100 v07 instructions. Modifications included DNA shearing with Covaris gTUBES (Covaris, Woburn, MA, USA). Library quality was assessed using the Agilent Femto Pulse system, followed by library quantification with the Qubit dsDNA High Sensitivity kit. The library was then sequenced on a PacBio Sequel II (PacBio, Menlo Park, CA, USA), using one SMRT cell and the Sequel Polymerase Binding kit 2.2 at the University of Wisconsin-Madison Biotechnology Center DNA Sequencing Facility.

The resulting raw PacBio reads were processed and filtered by CCS calling (CSS 6.2.2; https://github.com/PacificBiosciences/ccs), followed by demultiplexing. The demultiplexed reads were then assembled into contigs using hifiasm (https://hifiasm.readthedocs.io/en/latest/index.html#), a *de novo* PacBio HiFi read assembler [65]. The viral genomes were then manually assembled from contigs using Mega7 [66]. Following assembly, the genomes were annotated using VAPiD v1.3 [67].

## Multiple sequence alignment and recombination analysis

A multiple sequence alignment (MSA) for each OD4-CJ994 recombinant, including each parent, was separately generated using MAFFT v7.45 [68], with the FFT-NS-1 option. Recombination breakpoints for each individual recombinant were detected with RDP4 v4.101 [69], using 1,500 bp sliding window, a step-size of 500 bp, 300 bootstrap replicates, and the Jin and Nei nucleotide substitution model [70] options. A MSA including all 20 recombinants, the OD4 and CJ994 parental, and strain 17 reference sequence was also generated using MAFFT to map each of the breakpoints to strain 17 genome coordinates for data continuity.

## Ocular infection and disease scoring

Ocular infection and disease scoring has been described previously [63, 71–73]. Briefly, to assess the disease phenotype of the 20 novel OD4-CJ994 recombinants,10 females and 10 males (per recombinant), 4–6 week-old BALB/c mice underwent corneal scarification in the right eye with a 30-gauge needle, followed by placement of 1x10^5 PFU of virus in 5 µL of DMEM in 2% serum on the cornea. The eyes of the infected mice were examined 1, 3, 5, 7, 9, 11 and 13-days post-infection using a Wild-Heerbrugg M8 (Heerbrugg, SG, Switzerland) microscope. The blepharitis, neovascularization, and stromal keratitis ocular disease phenotypes were scored for severity as follows. Blepharitis: 1 +, mild swelling of eyelids; 2+, moderate swelling with some crusting; 3 +, eye swollen 50% shut with severe crusting; 4 +, eye crusted shut. Corneal neovascularization: 1+, less than 25% involvement (vessel ingrowth of the cornea from the limbus); 2+, 25 to 50% involvement; 3+, more than 50% involvement. Stromal

keratitis: 1 +, some haziness; iris detail visible; 2+, moderate clouding, iris detail obscured; 3+, cornea totally opaque; 4+, perforated cornea. For each of the ocular disease phenotypes, a graded score of 0 denotes no observable pathology.

For each viral recombinant, the mean peak disease scores (MPDS) of each of the three ocular disease phenotypes were separately determined according to biological sex by averaging the highest disease score for each animal during the 13-day study. Tear film samples were taken from each mouse by washing the eye with 10 μL DMEM + 2% serum (plus penicillin/streptomycin and amphotericin B), on days 1, 3, 5, and 7 post-infection and subsequently titered using serial dilutions on Vero cell monolayers. A separate average titer for each recombinant was calculated according to sex. Animal weights for each individual mouse were measured at baseline and on days 1, 3, 5, 7, 9, 11, and 13 post-infection. Food intake was not determined. The mean peak weight loss (MPWL) measure (average of the highest weight loss from baseline weight for each mouse over the course of the 13-day study) was calculated for each recombinant according to sex.

Signs of encephalitis, (neurovirulence) were also scored for each OD4-CJ994 recombinant on days 1, 3, 5, 7, 9, 11, and 13 post-infection. The scoring was binary under the categories of slow respiration rate, tremors/seizures, head tilt, circular gait, ataxia, paralysis, hyperactivity, and coma. A positive score in any of these categories resulted in an infected mouse being scored as positive for neurovirulence. The animal experiments observed the Association for Research in Vision and Ophthalmology and NIH animal welfare guidelines and were approved by the University of Wisconsin-Madison Institutional Animal Care and Use Committee (IACUC). Sustained release burprenorphine was used for analgesia and mice with severe neurological signs were humanely euthanized to prevent suffering.

## Statistical analysis

Animal ocular disease phenotypic data graphing, Mann-Whitney U test statistical analyses and regression analyses between phenotypes were performed using SigmaPlot 11 (Systat, Palo Alto, CA, USA). Linear regression comparison was performed using an operator (e.g. MPDS Blepharitis) as the response variable (the expected effect) and the independent variable (e.g. average percent neurovirulence; the experimental cause) interacted with "female" as the independent data. The P values associated with the female and variable: female coefficients indicate if there is a difference in the relationship between the variable and the operator (R software package v.3.6.1 [74]). For all statistical analyses, a P value of 0.001 or less was considered the threshold for significance.

## Results

We previously generated OD4/994 recombinants to map virulence determinants driving ocular infection in a mouse model of viral keratitis [54, 75]. However, we only scored disease phenotypes in female BALB/c mice. To determine if biological sex differences affected keratitis, we generated an additional set of recombinant viruses and quantified ocular virulence, neurovirulence, viral replication, and weight loss in both male and female BALB/c mice. This additional set of recombinants was generated to produce additional neurovirulent strains for an unrelated project.

To confirm the viruses were recombinants, the viral genomes were sequenced (See Table 1 for GenBank accession numbers), and parental sequences in each recombinant virus were identified using breakpoint analysis (Fig 1). Each virus was the result of multiple recombination events distributed across the genome with a range of 7 (INV-9B) to 24 (INV-17A) events per genome similar to what we reported previously [62].

**Table 1. Table of HSV-1 recombinant and parental genome lengths and GenBank accession numbers.**

| Recombinant Name | Genome Length | GenBank Accession Number |
|---|---|---|
| INV-2C | 155161 | OP871856 |
| INV-NN | 155472 | OP871852 |
| INV-1B | 156888 | OP871853 |
| INV-2B | 155681 | OP871855 |
| INV-16E | 159092 | OP871843 |
| INV-20C | 154107 | OP871846 |
| INV-9B | 156328 | OP871859 |
| INV-15B | 156713 | OP871841 |
| INV-22C | 153324 | OP871848 |
| INV-11A | 154353 | OP871840 |
| INV-7A | 153807 | OP871857 |
| INV-7E | 154992 | OP871858 |
| INV-17A | 154234 | OP871844 |
| INV-22B | 155163 | OP871847 |
| INV-20A | 154095 | OP871845 |
| INV-1G | 153201 | OP871854 |
| INV-16D | 155161 | OP871842 |
| INV-25B | 154157 | OP871849 |
| INV-25C | 153240 | OP871850 |
| INV-25D | 155512 | OP871851 |
| OD4 (parental) | 152011 | JN420342.2 |
| CJ994 (parental) | 152175 | KR011283.1 |

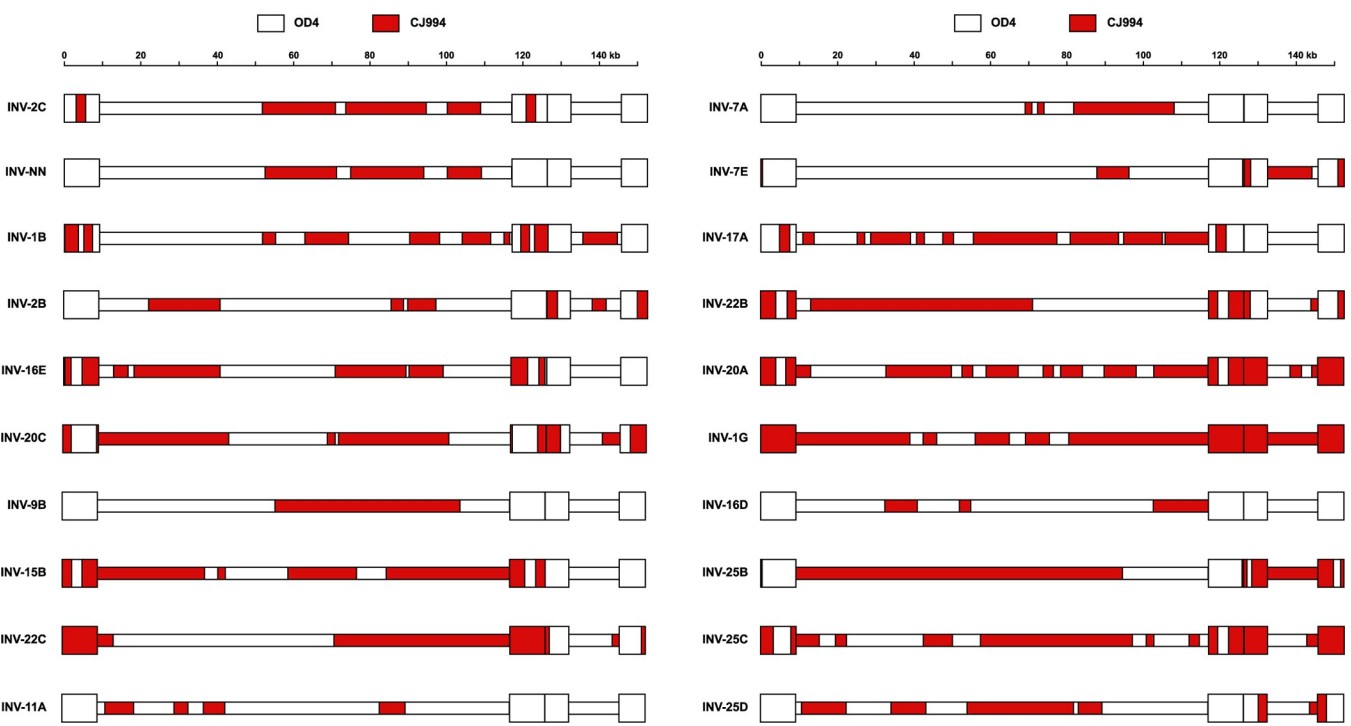

**Fig 1. Schematic representation of the parental contributions to each recombinant virus used in these studies.** Viral genome sequences from each parental virus were Identified and mapped to the standard HSV-1 Strain-17 genome (NC_001806.2). Red, CJ994, white OD4.

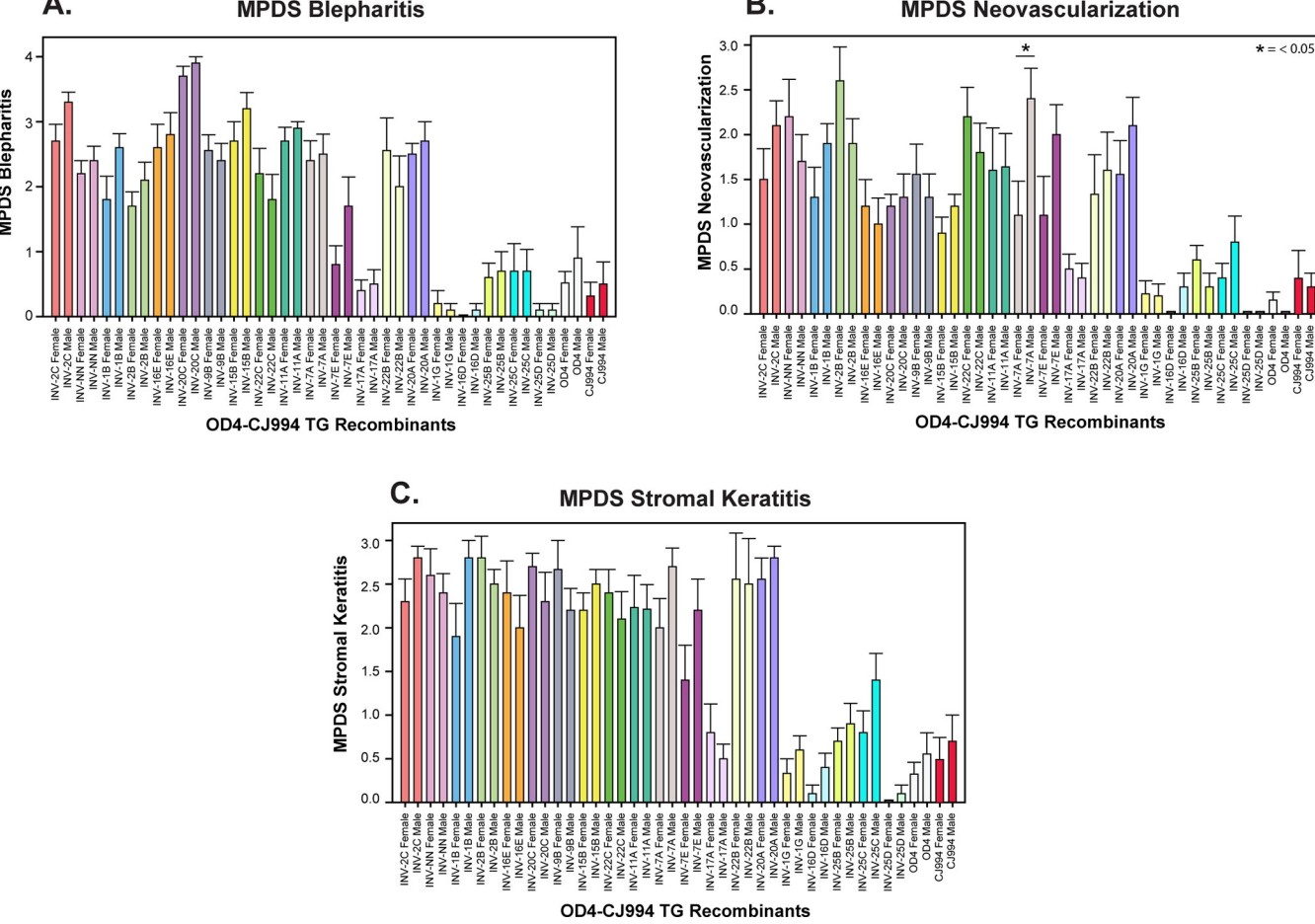

**Fig 2. Mean peak disease scores for ocular disease.** The peak disease scores for each mouse, regardless of the day, were averaged and the mean +/- standard error were plotted for each biological sex. The Mann-Whitney U test was used to assess significant differences between male and female mice for each recombinant virus. Panel A, blepharitis; Panel B, corneal neovascularization; Panel C, stromal disease (corneal clouding).

The parameters of ocular disease severity were scored for each sex with each recombinant virus and the mean peak disease scores (MPDS) are shown in Fig 2. As expected from our previous work [54, 75] the recombinants displayed varied disease phenotypes ranging from highly virulent to mild disease. For blepharitis and stromal disease, the differences in scores were not significantly different between sex for any of the recombinant viruses regardless of the disease severity. Only one recombinant, INV-7A (4.5%), showed significantly different neovascularization scores with male mice having more severe disease than females.

The percentage weight loss for each virus is shown in Fig 3. Weight loss was closely associated with virulence with between 20 and 25% loss for the most virulent viruses and under 5% for the low virulence viruses. Three recombinant viruses were identified as having significant sex differences for weight loss, INV-1B, INV-22C, and INV-11A. For INV-1B and INV-11A infected male mice lost more weight than female mice, while for mice infected with INV22C, female mice lost more weight than male mice so weight loss was not consistent across sexes and was only seen in 3/22 (13.6%) of the recombinants.

Peak eye wash titers are shown in Fig 4. There were fewer differences in terms of the peak titers for most of the viruses (approximately $10^4$ pfu) but was somewhat lower for 5 of the viruses ($10^3$ or less). Significant sex differences in titers on day 1 PI were observed with INV-

## Mean Peak Percent Weight Loss

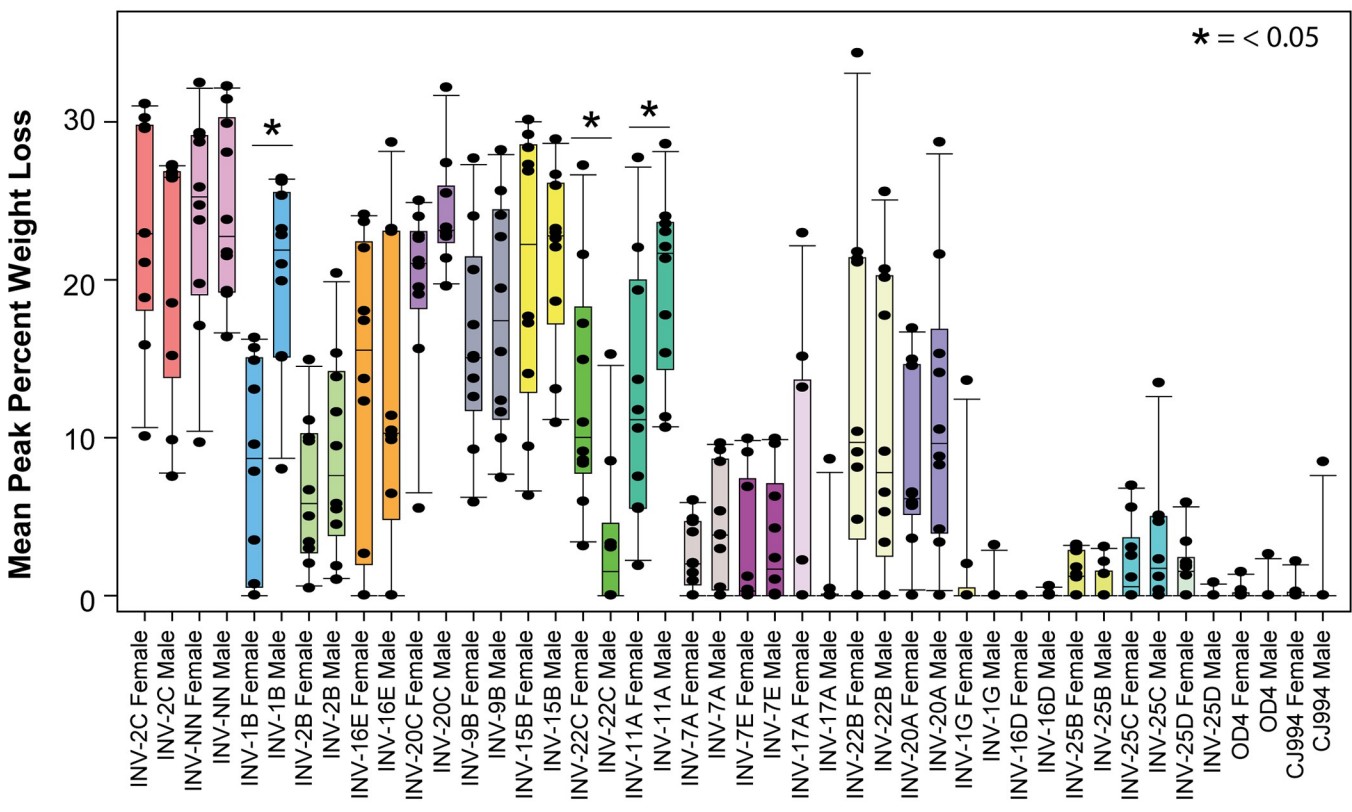

**Fig 3. Mean peak percent weight loss.** Mice were weighed on days 1, 3, 5, 7, 9, and 11 and the peak percent weight loss values, regardless of the day, were determined for each individual mouse. Box and whisker plots where generated for recombinant strain and biological sex. The dots superimposed onto the box plots are individual datapoints. The error bars of the plots denote the 10 and 90th percentile, the 25th and 75th percentile by the bottom and top of the box, and the median by the line within the box. The Mann-Whitney U test was used to determine if there were significant differences between male and female mice. Significant differences were identified for three recombinant viruses, INV-1B, INV-22C, and INV-11A but sex differences were not consistent as with some viruses male mice lost more weight while for others female mice lost more weight.

2C, INV-20C and INV-1G, with the titers from female mice being higher than males (S1 Table).

We also scored signs of neurological signs including tremors ataxia, hunched posture startle responses and malaise (Fig 5) and we found no significant differences between sexes for any recombinant virus.

To further evaluate sex differences, we carried out a regression analysis comparing the severity of ocular disease phenotypes separately for males and females. The results are shown in Fig 6 and S2 Table. Ocular disease severities separated into two groups: lower virulence recombinants with lower disease scores and higher virulence recombinants with higher disease scores. However, the $R^2$ values for males and females were nearly identical and not significantly different, indicating that correlations between virulence phenotypes do not depend on sex for ocular disease.

We also carried out regression analysis for neurovirulence against the ocular disease phenotypes and weight loss (Fig 7; S3 Table). The variability was higher than with the ocular disease comparisons and we did not find clustering of the low and high virulence viruses in these comparisons. The $R^2$ values for neurovirulence against blepharitis, weight loss, and corneal

## Mean Peak Titer

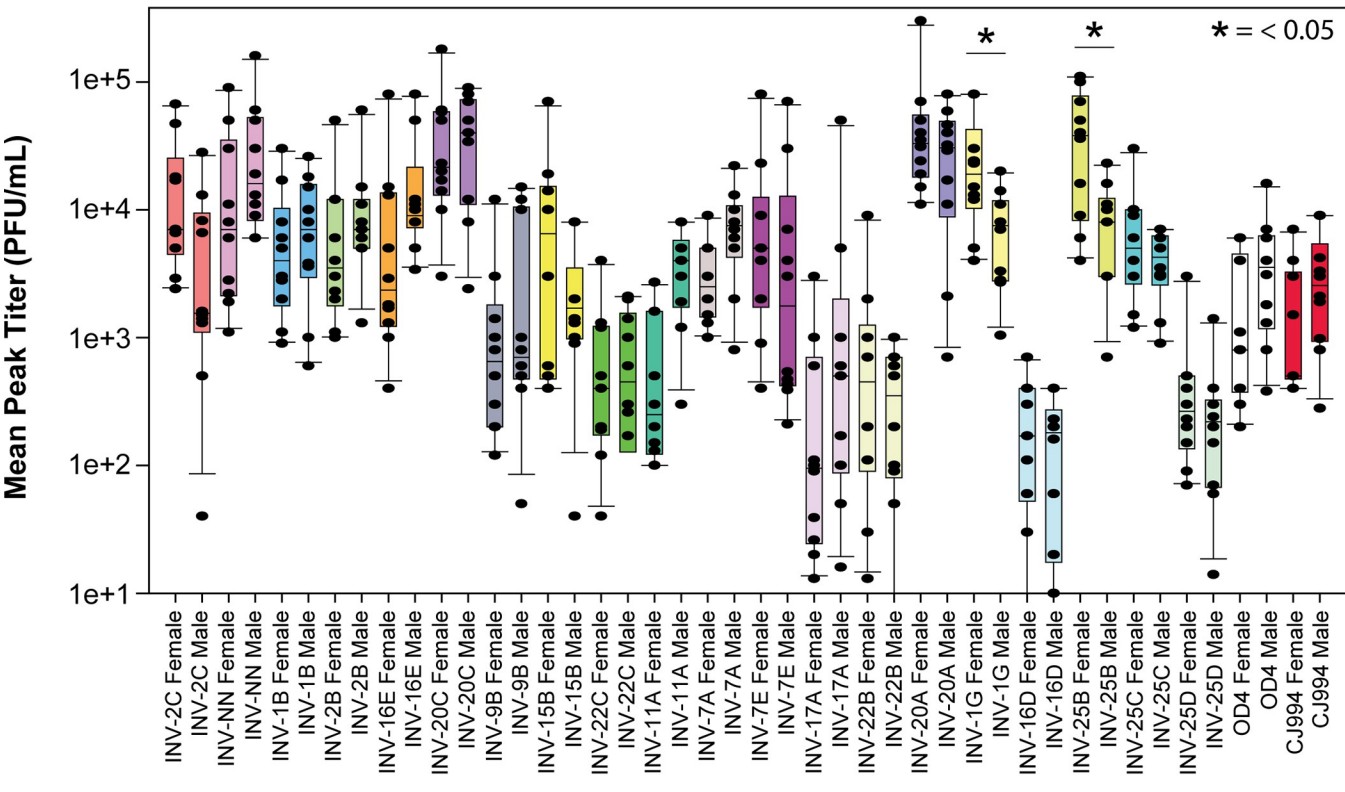

**Fig 4. Mean peak viral ocular titers.** Titer samples were collected on days 1, 3, 5, 7, and 9 and the peak titer values, regardless of the day were determined for each individual mouse. Box and whisker plots where generated for recombinant strain and biological sex. The dots superimposed onto the box plots are individual datapoints. The error bars of the plots denote the 10 and 90th percentile, the 25th and 75th percentile by the bottom and top of the box, and the median by the line within the box. The Mann-Whitney U test was used to evaluate significant differences. Significant differences between sexes were detected for INV-1G and INV-25B.

clouding ranged from 0.422 to 0.851, but as we found with the ocular disease score comparisons, the $R^2$ values for males and females were nearly identical. Interestingly, the $R^2$ values for neurovirulence vs. corneal neovascularization were low (approximately 0.24) indicating a lack of correlation. However, as with the other comparisons, the $R^2$ values for males and females were nearly identical.

## Discussion

Differences between biological sexes in incidence and disease severity for HSV are apparent for some clinical manifestations such as genital infection but do not seem to be involved in other forms of disease. For HSV-1 ocular infections, epidemiological studies in humans have either reported no sex differences in disease incidence or did not analyze sexes separately [40, 48–50]. Sex differences were not seen following ocular infection with HSV-1 strain McKrae in NIH/OLA mice [52]. A recent study using C57BL/6 mice infected with HSV-1 Strain 17 found that there were no sex differences in disease severity, viral titers, or immune infiltrates in the cornea [53] nor were consistent sex differences seen in a study using BXD mice [54]. The question of whether BALB/C mice would behave similarly or whether the virulence of the virus,

## Mean Positive Neurological Sign

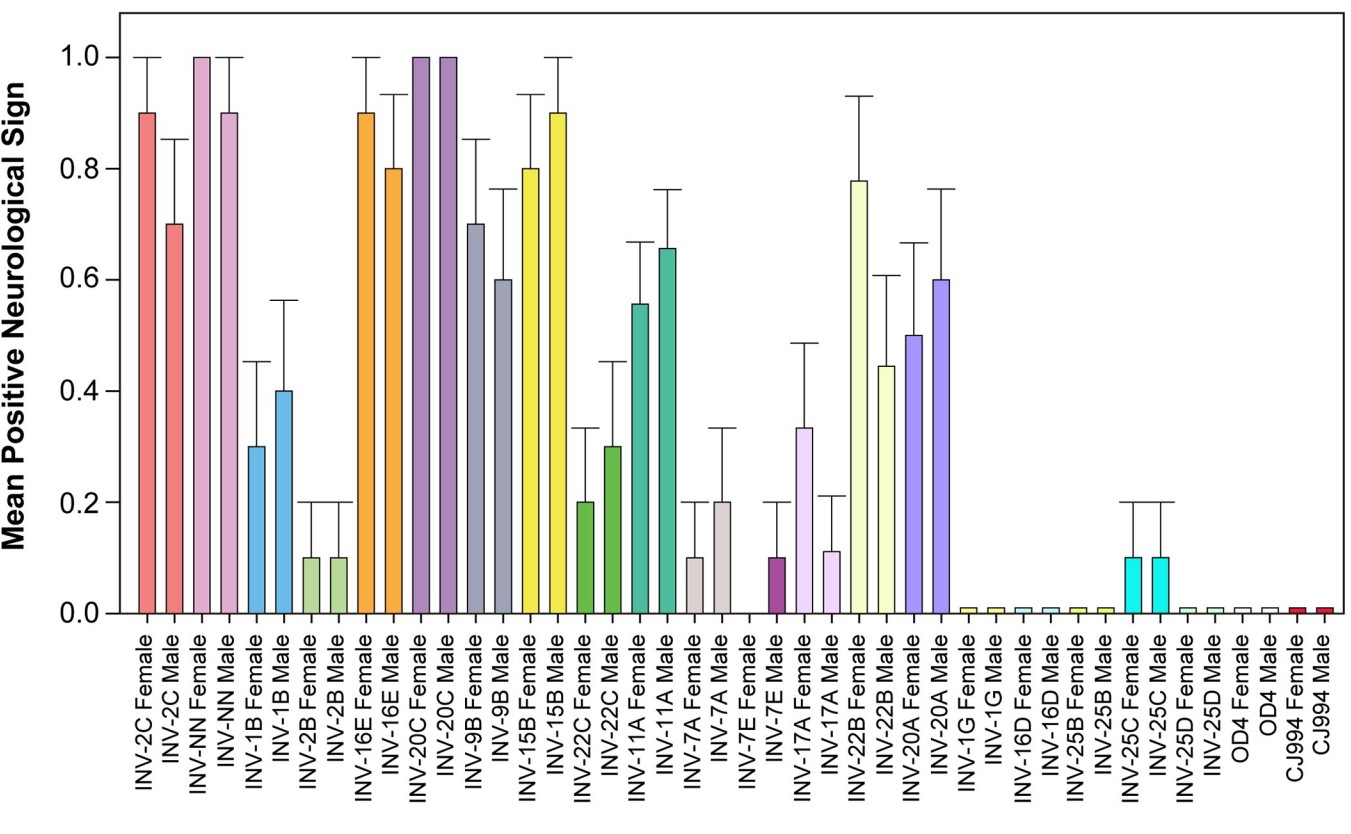

**Fig 5. Mean positive neurological signs.** Mice were evaluated on days 1, 3, 5, 7, and 9 for signs of neurological disease (encephalitis) including tremors, ataxia, hunched posture, startle responses, and malaise. Signs were scored either + or–and the values were averaged and the means +/- standard error were plotted. The Mann-Whitney U test was used to compare differences between male and female mice. There were no signficant differences between sexes for any of the recombinant viruses.

had in effect remained open questions. To that end, we generated a panel of viruses with different levels of virulence and compared several clinical parameters of ocular disease in male and female BALB/C mice. Our results are like those obtained with other strains of mice and in human studies.

For most disease parameters in our study, there were no sex differences, however, there was a sex difference for one virus for corneal neovascularization and for three viruses for weight loss. Although it is possible that each of these viruses has a genomic structure that leads to sex differences, our interpretation of the data is that, for ocular infection, sex differences that appeared to be significant are the result of variability in the disease model. There are several reasons for this. First, the viruses that showed a difference only had a difference in a single parameter and the differences were not consistent between males and females. Second, none of the viruses showed differences in more than a single outcome measure. Finally, a lack of sex difference would be consistent with epidemiological studies and studies in NIH/OLA, BXD and C57/BL6 mice [52–54]. Thus, our results extend the sex analysis to BALB/c mice and show that sex does not have an effect for ocular infection. Our data also show that sex does not affect the severity of disease caused by the infecting virus.

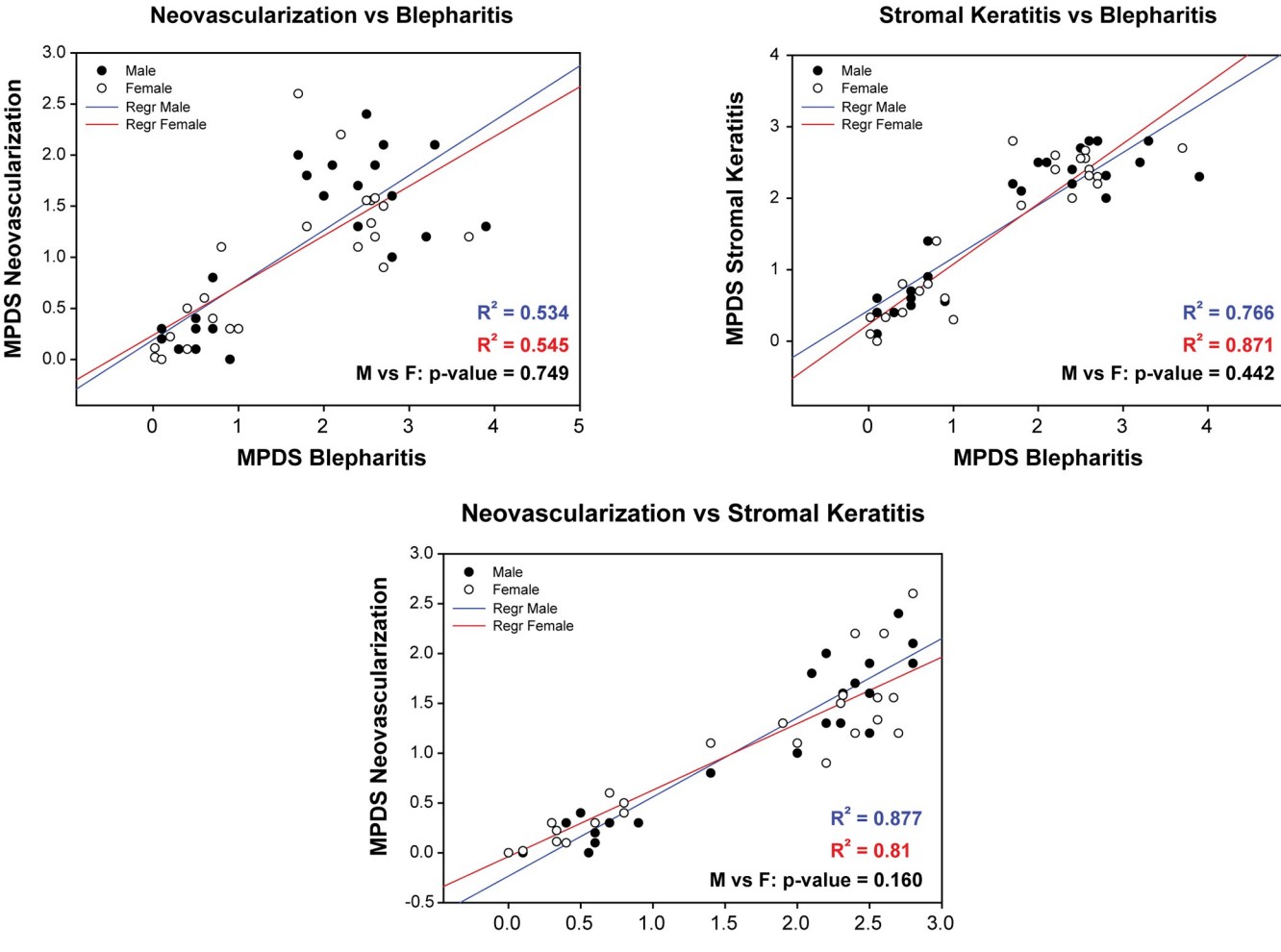

**Fig 6. Regression analysis comparing ocular disease parameters.** The MPDS scores were plotted against each other for each sex to determine potential correlations. Note that ocular disease scores clustered into 2 groups, viruses causing mild or no disease and those causing more severe disease. Note also that the $R^2$ values for the sexes were essentially identical. Closed circles, male mice; open circles, female mice. Blue, male mice; red, female mice.

The literature on the role of sex in herpes simplex virus infections is difficult to assess because of the use of multiple different models and the diversity of outcome measures that have been studied. Human studies for genital herpes (HSV-2) have shown the incidence is higher and symptoms are more severe in women than men [23]. It is not clear whether this is due to the different types of tissues infected (keratinized skin in men and mucosal epithelium in women) or to other factors such as differences in immune function due to steroid hormones. Hill et al. [38] found that the copy number of HSV-1 DNA in human trigeminal ganglia did not differ by sex, suggesting there are no differences in viral trafficking to the trigeminal ganglia and establishment of latency. Other studies have reported mixed results for sex differences such that no clear pattern of susceptibility and resistance between males and females is evident [31, 76, 77]. Liesegang et al. [40], reported that there was no statistically significant difference between men and women in the incidence of herpes simplex keratitis, but disease severity was not reported.

Studies using mouse models have also given mixed results. Some report that females are more susceptible or have more severe outcomes while others suggest that male mice are more

## Weight Loss vs Neurovirulence

## Blepharitis vs Neurovirulence

## Neovascularization vs Neurovirulence

## Stromal Keratitis vs Neurovirulence

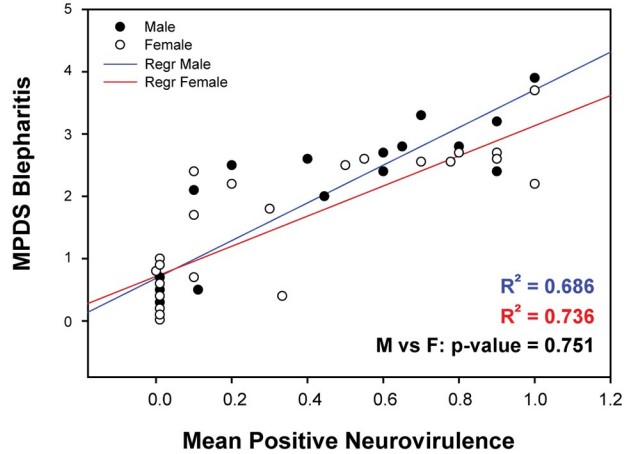
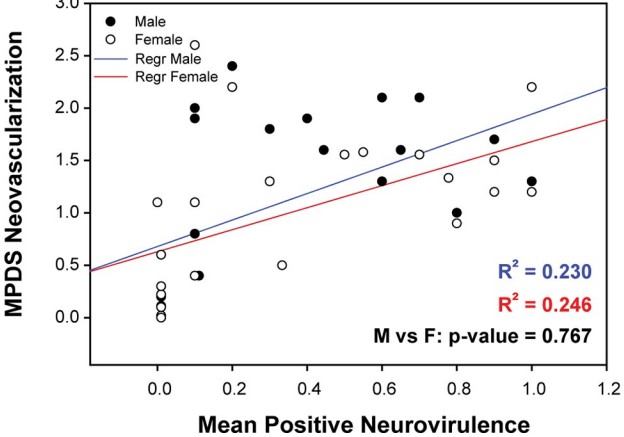
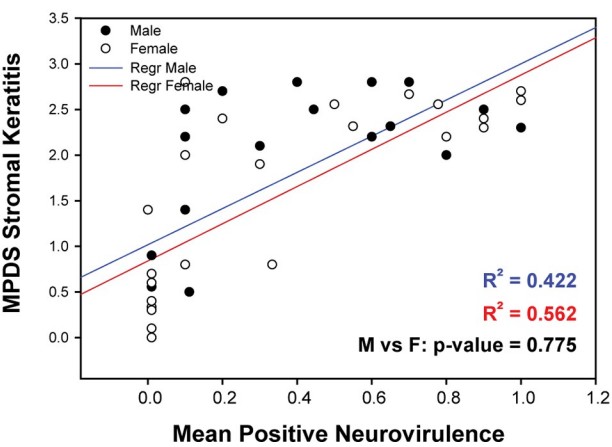

**Fig 7. Regression analysis of ocular disease scores and weight loss compared to neurovirulence.** The mean peak disease scores for blepharitis, neovascularization, stromal disease, and weight loss were plotted against the neurovirulence scores. Correlations were identified for blepharitis, weight loss and stromal disease but not for neovascularization and neurovirulence. Closed circles, male mice; open circles, female mice. Blue, male mice; red, female mice. Note that the $R^2$ values for male and female mice were virtually identical.

susceptible or get more severe outcomes [30, 32, 33, 35]. Susceptibility gene mapping identified QTLs on multiple chromosomes that differentially contributed to infection in males and females [78]. Lundberg et al. [79] identified a locus, termed *Sml*, that augmented resistance of female mice to HSV-1 infection (not ocular). Sivasubramanian et. al. [32] reported that neuroinflammation and senescence was enhanced in the brainstem of female C57/BL6 mice, but the virus used was a LAT -/- mutant.

A recent paper [54] used a panel (BXD) of recombinant mice to map genetic loci affecting the severity of disease in an ocular infection model. The phenotypic data that were reported included weight loss, viral titers in the eye, keratitis severity, and death from encephalitis. They found that for some of the lines (e.g., BXD34) there was a sex difference for weight loss but to quote; "A sex difference in susceptibility was not universal". A locus on chromosome 12 was involved in weight loss with both sexes. Viral titers in at least 4 lines of mice were not different between sexes. Sex differences for encephalitis were not reported. Finally, they scored the

severity of keratitis, but did not report sex differences. In our study any sex differences in clinical parameters were also not universal and only appeared in a small number of the viral recombinants that were tested. More recently, Moein et. al. [51] included mice of both sex but sexes were apparently not analyzed separately.

We previously used vQTLmap to identify HSV-1 genes associated with ocular disease severity [62, 75] and found the predominant genes involved were regulatory or were genes that were known to counter immune defenses. To date, we have not analyzed neurovirulence and were not able to do so for this set of recombinants due to the small sample size but this is planned for future work.

## Concluding remarks

Many studies of biological sex differences in Herpes simplex virus infections have focused on immune responses, and they have shown differences at the cellular and molecular level, however, the differences are not consistent between males or females and depend on the virus and disease being considered. Depending on the outcome measure, males sometimes have stronger responses while females have stronger responses for other outcomes. Given that HSV-1 keratitis is an immunopathological disease, with pro-inflammatory cytokine production and corneal infiltration by neutrophils and CD4+ T-cells, one might have expected to find that sex differences in immune responses would affect disease severity. However, our results and the results of Shimeld et al. [52], Riccio et al. [53] and Thompson et al. [54] show that at the level of clinical disease severity, viral titers, neurovirulence, and weight loss, sex differences were not apparent. If there are sex differences in immune responses at the cellular and molecular level, they do not translate to the organismal level in terms of disease severity in mouse ocular models. Therefore, for some uses, murine disease models including both sexes should not be necessary.

## Supporting information

**S1 Table. Average ocular titers on days 1, 3, 5, and 7 days post-infection from male and female BALB/c mice for each viral recombinant.**
(XLSX)

**S2 Table. Regression analysis detailed results comparing the severity of HSV-1 disease phenotypes between male and female mice.**
(XLSX)

**S3 Table. Regression analysis comparison results for HSV-1 disease phenotypes between male and female mice.**
(XLSX)

## Acknowledgments

The authors thank Dr. Donna Neumann for helpful comments on the manuscript.

## Author Contributions

**Conceptualization:** Aaron W. Kolb, Curtis R. Brandt.

**Funding acquisition:** Curtis R. Brandt.

**Investigation:** Aaron W. Kolb, Sarah A. Ferguson, Inna V. Larsen, Curtis R. Brandt.

**Methodology:** Aaron W. Kolb, Curtis R. Brandt.

Writing – **original draft:** Curtis R. Brandt.

Writing – **review & editing:** Aaron W. Kolb, Sarah A. Ferguson, Inna V. Larsen, Curtis R. Brandt.

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
