## [Decision Letter · Decision Letter 0]

13 Apr 2023

PONE-D-23-06338Disease Parameters Following Ocular Herpes Simplex Virus Type 1 Infection are Similar in Male and Female BALB/C MicePLOS ONE

Dear Dr. Brandt,

Thank you for submitting your manuscript to PLOS ONE. After careful consideration, we feel that it has merit but does not fully meet PLOS ONE’s publication criteria as it currently stands. Therefore, we invite you to submit a revised version of the manuscript that addresses the points raised during the review process.

We look forward to receiving your revised manuscript.

Kind regards,

Homayon Ghiasi, PhD

Academic Editor

PLOS ONE

Journal Requirements:

Reviewers' comments:

Reviewer's Responses to Questions

**Comments to the Author**

1. Is the manuscript technically sound, and do the data support the conclusions?

Reviewer #1: Yes

Reviewer #2: Yes

2. Has the statistical analysis been performed appropriately and rigorously? 

Reviewer #1: Yes

Reviewer #2: Yes

3. Have the authors made all data underlying the findings in their manuscript fully available?

Reviewer #1: Yes

Reviewer #2: Yes

4. Is the manuscript presented in an intelligible fashion and written in standard English?

Reviewer #1: Yes

Reviewer #2: Yes

5. Review Comments to the Author

Reviewer #1: Some studies regarding herpes simplex virus 1 (HSV-1) and 2 (HSV-2) have indicated that aspects of infections and disease presentations show significant differences between males and females; other studies indicate that there are no noticeable sex differences in pathogenesis between HSV-1 or HSV-2. The goal of the current study is for the authors to examine differences in viral replication and signs of disease based on sex using an ocular model of HSV-1 infection in BALB/C mice. Mice in this study were infected with recombinant viruses derived from 2 different strains of HSV-1. The authors show that there were no overall sex-based differences in ocular diseases and viral replication, as consistent phenotypes were not observed among either male or female mice for all recombinants tested. The authors data indicate that use of both male and female mice is not necessary for most HSV-1 BALB/C studies. The data presented is fairly straightforward and interpretable. Further clarity related to one of the premises in the manuscripts, as well as the significance of data in supplementary dat in Tables 1 and 2 would significantly strengthen the manuscript. Specific points are noted below.

1. Page 4, lines 14-16: “Two critical questions….whether sex differences exist……in BALB/c mice…” Based on the background provided in the paragraph, focus on BALB/c mice as one of the authors “critical” questions is not apparent. Please provide a clear rationale as why the use of BALB/c mice is important for this study.

2. From the data presented in Supplementary Data presented in Tables 1 and 2, it is unclear as to what specific points or interpretations the authors would like the reader to take away from the data compared to straightforward regression analyses and graphs presented in Figures 6 and 7. In other words, the data presented in the tables do not provide an intuitive understanding of the authors’ results compared to the aforementioned figures. Please clearly indicate in the Tables 1 and 2 what specific points in the regression/statistical analyses support the authors results or conclusions.

3. Page 5, first line: “…..sex differences for are not….” This part of this sentence is incomplete. Please edit this sentence to make it complete.

Reviewer #2: The authors present a comprehensive analysis of 20 new recombinants of two attenuated strains (OD4 and CJ994) of herpes simplex virus 1 (HSV-1). This work follows on prior papers from this lab, which used an earlier set of viral recombinants of the same strains to map viral genetic regions that may contribute to different aspects of virulence in a murine ocular model of infection. Here the authors apply similar methods, but ask the question of whether the sex of the mice influences their response to HSV-1 infection – at least in the BALB/C background examined here. The overall answer is that sex is not a major factor in BALB/C mouse response to infection, at least for these recombinants and at this dose and murine age. There are minor effects of sex, which leaves the door open for future studies to probe this more deeply.

Overall there are no technical flaws or issues to correct in the study. The data are well-explained, and figures are clear. The text was a pleasure to read and covers all the areas of interest. These suggestions are mainly to add context or rationale in a few areas of interest.

- Was there any reason to carry out this study with a new set of 20 OD4-CJ994 recombinants, instead of any of the many prior recombinants that have already been published on? This is just a point of curiosity that other readers will likely share. A simple line of comment on this would be insightful for readers.

- The histograms in Figures 2-5 are color-coordinated and enable easy comparison across figures. Was there any reason to present the recombinants in the left-to-right order they are in? It might be helpful to rank order them according to the virulence-category with the greatest variance, and then apply that order to all figures. That would make it easier to see the more-virulent vs. less-virulent recombinants. At present the “attenuated” ones are somewhat clustered on the right in Figure 2, but have a few attenuated ones off the left.

- Was there any attempt to check the replication ability of these recombinants in cell culture, particularly for the ones with low replication in tear films? It seems likely that any in vitro growth defects would have been picked up in the process of plaque-purification and/or viral stock or DNA production, but at present this isn’t mentioned in the text.

- Is there any context that can be added from prior literature about sex differences in BALB/c mice relative to other murine backgrounds? E.g., There is a sizable lifespan difference for males vs. females in BALB/c mice – to what extent is this similar to that of other common murine models used to study HSV-1 pathogenesis? Do the authors infer that these results would likely be similar if they or another group repeated this study in a different murine background?

- Similarly, do the authors anticipate that the responses would be similar if they applied a different viral dose, or used mice of a different age?

Minor note:

- There are several instances where it appears that incorrect references are pointed to. Please check and correct these where needed. E.g., on page 1, this cite [2,23-28] -- #2 is ~ influenza and likely not the intended citation. Same page, this cite [13,30] -- #30 is a murine study and likely not the intended (human-related) citation; on page 30, this cite [50] should likely be #51 instead.

6. PLOS authors have the option to publish the peer review history of their article (what does this mean?). If published, this will include your full peer review and any attached files.

Reviewer #1: No

Reviewer #2: No

---

## [Author Response · Author response to Decision Letter 0]

22 May 2023

Please thank the reviewers for their thoughtful comments. We have responded by making some changes in the manuscript that are detailed below. Some of the comments referred to issues that are required by PLOS ONE for publication or may make the data more difficult to interpret. Additionally, in the “data not show” on page on page 12 has been removed as per PLOS ONE requirements, and a supplementary table containing ocular titers per day has been added.

Reviewer 1

The reviewer’s first comment related to the importance of BALB/c mice for this study and felt it was not clearly defined. We have added text in the introduction which addresses this issue, specifically that there are phenotypic differences with regards to herpes ocular disease severity between mouse strains and that BALB/c mice are one of the most common mouse strains used in herpes ocular studies. 

The reviewer was unclear as to the purpose of the statistical data presented in supplementary tables 1 and 2. These tables are present in the manuscript to meet PLOS ONE’s statistical reporting guidelines (https://journals.plos.org/plosone/s/submission-guidelines.#loc-statistical-reporting). 

The reviewer also found an incomplete sentence found on page 5. Text has been added to address this issue.

Reviewer 2 

The reviewer first questioned why additional recombinants were generated rather than using previously published ones. The additional recombinants presented in the manuscript were generated for an unrelated project, and additional text has been added to reflect this. 

The reviewer wondered if there was a reason recombinants were in a certain order in the histograms presented in Figures 2-5, and posited that we should change them, for example in order of greatest variation. We have declined to do so as we believe presenting the data this way would result in significantly increased difficulty interpreting the data. For example, strains showing the most variability in blepharitis, would not necessarily be variable in other phenotypes such as neovascularization. 

The reviewer asked if there was any attempt to check the replication ability of the recombinants in cell culture. We have not done so as the focus of the manuscript is looking for possible biological sex differences with respect to viral strain, not differences between the strains themselves. This would also require identifying a cell line where replication differences would likely be significant, and running the assays would likely take longer than the time allotted to return the manuscript.

The reviewer next asked if there are any prior studies regarding sex differences in BALB/c mice relative to other backgrounds and wonder to the extent of these differences to other mouse strains used in HSV studies. There have been past studies looking into sex and murine strain differences, for example airway inflammation (doi: 10.1038/s41598-022-25327-7), pain (doi: 10.30802/AALAS-CM-19-000066) and social behavior (doi: 10.1016/j.isci.2022.103735). While differences exist between sexes and mouse strains, it is difficult to predict how these may impact HSV-1 ocular disease. Related to this the reviewer asked how the results may differ if different viral doses or mouse ages were used. Different viral inoculums are unlikely to have a significant effect, as previous work has shown that keratitis and neurovirulence severity are not titer dependent (Invest Ophthalmol Vis Sci 30:2474-2480,1989). While it’s possible age related differences between sexes may exist, it not possible to address this in the time given to return the manuscript. 

The reviewer also noticed a couple incorrect references and those have been addressed.

---

## [Editor Report · Decision Letter 1]

1 Jun 2023

Disease Parameters Following Ocular Herpes Simplex Virus Type 1 Infection are Similar in Male and Female BALB/C Mice

PONE-D-23-06338R1

Dear Dr. Brandt:

We’re pleased to inform you that your manuscript has been judged scientifically suitable for publication and will be formally accepted for publication once it meets all outstanding technical requirements.

Kind regards,

Homayon Ghiasi, PhD

Academic Editor

PLOS ONE
---

## [Editor Report · Acceptance letter]

8 Jun 2023

PONE-D-23-06338R1 

Disease Parameters Following Ocular Herpes Simplex Virus Type 1 Infection are Similar in Male and Female BALB/C Mice 

Dear Dr. Brandt:

I'm pleased to inform you that your manuscript has been deemed suitable for publication in PLOS ONE. Congratulations! Your manuscript is now with our production department. 

Kind regards, 

on behalf of

Dr. Homayon Ghiasi 

Academic Editor

PLOS ONE